# DYNAMIC GRAPH REPRESENTATION LEARNING VIA SELF-ATTENTION NETWORKS

## ABSTRACT

Learning latent representations of nodes in graphs is an important and ubiquitous task with widespread applications such as link prediction, node classification, and graph visualization. Previous methods on graph representation learning mainly focus on static graphs, however, many real-world graphs are dynamic and evolve over time. In this paper, we present Dynamic Self-Attention Network (DySAT), a novel neural architecture that operates on dynamic graphs and learns node representations that capture both structural properties and temporal evolutionary patterns. Specifically, DySAT computes node representations by jointly employing self-attention layers along two dimensions: structural neighborhood and temporal dynamics. We conduct link prediction experiments on two classes of graphs: communication networks and bipartite rating networks. Our experimental results show that DySAT has a significant performance gain over several different state-of-the-art graph embedding baselines.

## 1 INTRODUCTION

Learning latent representations (or embeddings) of nodes in graphs has been recognized as a fundamental learning problem due to its widespread use in various domains such as social media (Perozzi et al., 2014), biology (Grover & Leskovec, 2016), and knowledge bases (Wang et al., 2014). The basic idea is to learn a low-dimensional vector for each node, which encodes the structural properties of a node and its neighborhood (and possibly attributes). Such low-dimensional representations can benefit a plethora of graph analytical tasks such as node classification, link prediction, and graph visualization (Perozzi et al., 2014; Tang et al., 2015; Grover & Leskovec, 2016; Wang et al., 2016).

Previous work on graph representation learning mainly focuses on static graphs, which contain a fixed set of nodes and edges. However, many graphs in real-world applications are intrinsically dynamic, in which graph structures can evolve over time. They are usually represented as a sequence of graph snapshots from different time steps (Leskovec et al., 2007). Examples include academic co-authorship networks where authors may periodically switch their collaboration behaviors and email communication networks whose structures may change dramatically due to sudden events. In such scenarios, modeling temporal evolutionary patterns is important in accurately predicting node properties and future links.

Learning dynamic node representations is challenging, compared to static settings, due to the complex time-varying graph structures: nodes can emerge and leave, links can appear and disappear, and communities can merge and split. This requires the learned embeddings not only to preserve structural proximity of nodes, but also to jointly capture the temporal dependencies over time. Though some recent work attempts to learn node representations in dynamic graphs, they mainly impose a temporal regularizer to enforce smoothness of the node representations from adjacent snapshots (Zhu et al., 2016; Li et al., 2017; Zhou et al., 2018). However, these approaches may fail when nodes exhibit significantly distinct evolutionary behaviors. Trivedi et al. (2017) employ a recurrent neural architecture for temporal reasoning in multi-relational knowledge graphs. However, their temporal node representations are limited to modeling first-order proximity, while ignoring the structure of higher-order graph neighborhoods.

Attention mechanisms have recently achieved great success in many sequential learning tasks such as machine translation (Bahdanau et al., 2015) and reading comprehension (Yu et al., 2018). The

key underlying principle is to learn a function that aggregates a variable-sized input, while focusing on the parts most relevant to a certain context. When the attention mechanism uses a single sequence as both the inputs and the context, it is often called *self-attention*. Though attention mechanisms were initially designed to facilitate Recurrent Neural Networks (RNNs) to capture long-term dependencies, recent work by Vaswani et al. (2017) demonstrates that a fully self-attentional network itself can achieve state-of-the-art performance in machine translation tasks. Velickovic et al. (2018) extend self-attention to graphs by enabling each node to attend over its neighbors, achieving state-of-the-art results for semi-supervised node classification tasks in static graphs.

As dynamic graphs usually include periodical patterns such as recurrent links or communities, attention mechanisms are capable of utilizing information about most relevant historical context, to facilitate future prediction. Inspired by recent work on attention techniques, we present a novel neural architecture named Dynamic Self-Attention Network (DySAT) to learn node representations on dynamic graphs. Specifically, we employ self-attention along two dimensions: structural neighborhoods and temporal dynamics, *i.e.*, DySAT generates a dynamic representation for a node by considering both its neighbors and historical representations, following a self-attentional strategy. Unlike static graph embedding methods that focus entirely on preserving structural proximity, we learn dynamic node representations that reflect the temporal evolution of graph structure over a varying number of historical snapshots. In contrast to temporal smoothness-based methods, DySAT learns attention weights that capture temporal dependencies at a fine-grained node-level granularity.

We evaluate our framework on the dynamic link prediction task using four benchmarks of different sizes including two email communication networks (Klimt & Yang, 2004; Panzarasa et al., 2009) and two bipartite rating networks (Harper & Konstan, 2016). Our evaluation results show that DySAT achieves significant improvements (3.6% macro-AUC on average) over several state-of-the-art baselines and maintains a more stable performance over different time steps.

## 2 RELATED WORK

Our framework is related to previous representation learning techniques on static graphs, dynamic graphs, and recent developments in self-attention mechanisms.

**Static graph embeddings.** Early work on unsupervised graph representation learning exploits the spectral properties of various graph matrix representations, such as Laplacian, etc. to perform dimensionality reduction (Tenenbaum et al., 2000; Belkin & Niyogi, 2001). To improve scalability, some work (Perozzi et al., 2014; Grover & Leskovec, 2016) utilizes Skip-gram methods, inspired by their success in Natural Language Processing (NLP). Recently, several graph neural network architectures based on generalizations of convolutions have achieved tremendous success, among which many methods are designed for supervised or semi-supervised learning tasks (Niepert et al., 2016; Defferrard et al., 2016; Kipf & Welling, 2017; Sankar et al., 2017; Velickovic et al., 2018). Hamilton et al. (2017b) extend graph convolutional methods through trainable neighborhood aggregation functions, to propose a general framework applicable to unsupervised representation learning. However, these methods are not designed to model temporal evolutionary patterns in dynamic graphs.

**Dynamic graph embeddings.** Most techniques employ temporal smoothness regularization to ensure embedding stability across consecutive time-steps (Zhu et al., 2016). Zhou et al. (2018) additionally use triadic closure (Kossinets & Watts, 2006) as guidance, leading to significant improvements. Neural methods were recently explored in the knowledge graph domain by Trivedi et al. (2017), who employ a recurrent neural architecture for temporal reasoning. However, their model is limited to tracing link evolution, thus limited to capturing first-order proximity. Goyal et al. (2017) learn incremental node embeddings through initialization from the previous time steps, however, this may not guarantee the model to capture long-term graph similarity. A few recent works (Nguyen et al., 2018; Zuo et al., 2018) examine a related setting of temporal graphs with continuous time-stamped links for representation learning, which is however orthogonal to the established problem setup of using dynamic graph snapshots. Li et al. (2017) learn node embeddings in dynamic attributed graphs by initially training an offline model, followed by incremental updates over time. However, their key focus is online learning to improve efficiency over re-training static models, while our goal is to improve representation quality by exploiting the temporal evolutionary patterns in graph structure. Unlike previous approaches, our framework captures the most relevant historical contexts through a self-attentional architecture, to learn dynamic node representations.

**Self-attention mechanisms.** Recent advancements in many NLP tasks have demonstrated the superiority of *self-attention* in achieving state-of-the-art performance (Vaswani et al., 2017; Lin et al., 2017; Tan et al., 2018; Shen et al., 2018; Shaw et al., 2018). In DySAT, we employ self-attention mechanisms to compute a dynamic node representation by attending over its neighbors and previous historical representations. Our approach of using self-attention over neighbors is closely related to the Graph Attention Network (GAT) (Velickovic et al., 2018), which employs neighborhood attention for semi-supervised node classification in a static graph. As dynamic graphs usually contain periodical patterns, we extend the self-attention mechanisms over the historical representations of a particular node to capture its temporal evolution behaviors.

## 3 PROBLEM DEFINITION

In this work, we address the problem of dynamic graph representation learning. A dynamic graph is defined as a series of observed snapshots, $\mathbb{G} = \{\mathcal{G}^1, \ldots, \mathcal{G}^T\}$ where $T$ is the number of time steps. Each snapshot $\mathcal{G}_t = (\mathcal{V}, \mathcal{E}^t)$ is a weighted undirected graph with a shared node set $\mathcal{V}$, a link set $\mathcal{E}^t$, and weighted adjacency matrix $\boldsymbol{A}^t$ at time $t$. Unlike some previous work that assumes links can only be added over time in dynamic works, we also allow to remove links. Dynamic graph representation learning aims to learn latent representations $\boldsymbol{e}_v^t \in \mathbb{R}^d$ for each node $v \in \mathcal{V}$ at time steps $t = 1, 2, \ldots, T$, such that $\boldsymbol{e}_v^t$ preserves both the local graph structures centered at $v$ and its evolutionary behaviors prior to time $t$.

## 4 DYNAMIC SELF-ATTENTION NETWORK

In this section, we first describe the high-level structure of our model. DySAT consists of two major novel components: *structural* and *temporal self-attention* layers, which can be utilized to construct arbitrary graph neural architectures through stacking of layers. Similar to existing studies on attention mechanisms, we employ multi-head attentions to improve model capacity and stability.

DySAT consists of a *structural* block followed by a *temporal* block, as illustrated in Figure 1, where each block may contain multiple stacked layers of the corresponding layer type. The *structural* block extracts features from the local neighborhood through self-attentional aggregation, to compute intermediate node representations for each snapshot. These representations feed as input to the *temporal* block, which attends over multiple time steps, capturing temporal variations in the graph.

### 4.1 STRUCTURAL SELF-ATTENTION

The input of this layer is a graph snapshot $\mathcal{G} \in \mathbb{G}$ and a set of input node representations $\{\boldsymbol{x}_v \in \mathbb{R}^D, \forall v \in \mathcal{V}\}$ where $D$ is the input embedding dimension. The input to the initial layer can be set as 1-hot encoded vectors for each node (or attributes if available). The output is a new set of node representations $\{\boldsymbol{z}_v \in \mathbb{R}^F, \forall v \in \mathcal{V}\}$ with $F$ dimensions that capture local structural properties.

Specifically, the *structural* self-attention layer attends over the immediate neighbors of a node $v$ (in snapshot $\mathcal{G}$), by computing attention weights as a function of their input node embeddings. The structural attention layer is a variant of GAT (Velickovic et al., 2018), applied on a single snapshot:

$$\boldsymbol{z}_v = \sigma\Big( \sum_{u \in \mathcal{N}_v} \alpha_{uv} \boldsymbol{W}^s \boldsymbol{x}_u \Big), \quad \alpha_{uv} = \frac{\exp\Big( \sigma\Big( A_{uv} \cdot \boldsymbol{a}^T [\boldsymbol{W}^s \boldsymbol{x}_u || \boldsymbol{W}^s \boldsymbol{x}_v] \Big) \Big)}{\sum\limits_{w \in \mathcal{N}_v} \exp\Big( \sigma\Big( A_{wv} \cdot \boldsymbol{a}^T [\boldsymbol{W}^s \boldsymbol{x}_w || \boldsymbol{W}^s \boldsymbol{x}_v] \Big) \Big)} \quad (1)$$

where $\mathcal{N}_v = \{u \in \mathcal{V} : (u, v) \in \mathcal{E}\}$ is the set of immediate neighbors of node $v$ in snapshot $\mathcal{G}$; $\boldsymbol{W}^s \in \mathbb{R}^{D \times F}$ is a shared weight transformation applied to each node in the graph; $\boldsymbol{a} \in \mathbb{R}^{2D}$ is a weight vector parameterizing the attention function implemented as feed-forward layer; $||$ is the concatenation operation and $\sigma(\cdot)$ is a non-linear activation function. Note that $A_{uv}$ is the weight of link $(u, v)$ in the current snapshot $\mathcal{G}$. The set of learned coefficients $\alpha_{uv}$, obtained by a softmax over the neighbors of each node, indicate the importance or contribution of node $u$ to node $v$ at the current snapshot. We use a LeakyRELU non-linearity to compute the attention weights, followed by ELU for the output representations. In our experiments, we employ sparse matrices to implement the *masked* self-attention over neighbors.

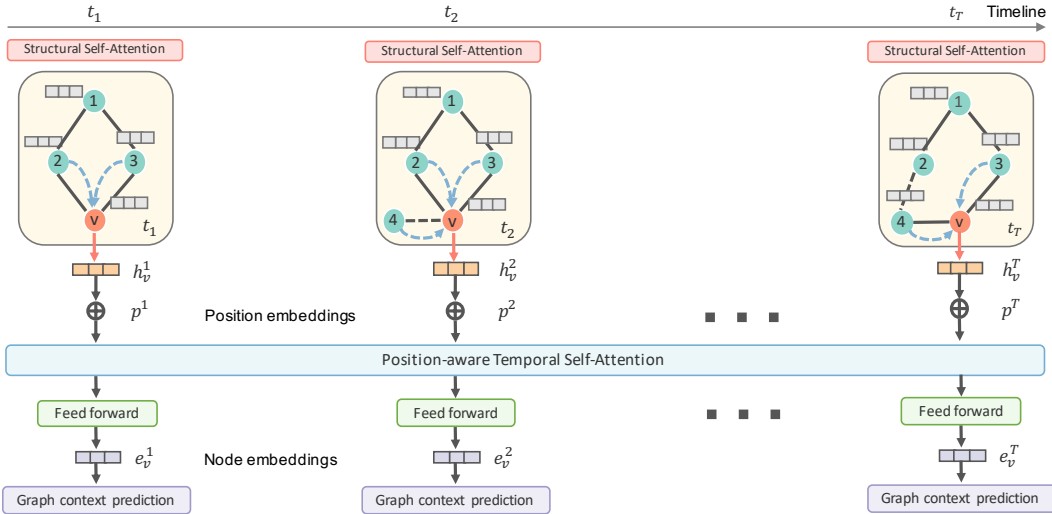

Figure 1: Neural architecture of DySAT: we employ structural attention layers followed by temporal attention layers. Dashed black arrows indicate new links and dashed blue arrows refer to neighbor-based structural-attention.

## 4.2 TEMPORAL SELF-ATTENTION

To further capture temporal evolutionary patterns in a dynamic network, we design a temporal self-attention layer. The input of this layer is a sequence of representations of a particular node $v$ at different time steps. Specifically, for each node $v$, we define the input as $\{\boldsymbol{x}_v^1, \boldsymbol{x}_v^2, \ldots, \boldsymbol{x}_v^T\}, \boldsymbol{x}_v^t \in \mathbb{R}^{D'}$ where $T$ is the number of time steps and $D'$ is the dimensionality of the input representations. The layer output is a new representation sequence for $v$ at each time step, $i.e.$, $\boldsymbol{z}_v = \{\boldsymbol{z}_v^1, \boldsymbol{z}_v^2, \ldots, \boldsymbol{z}_v^T\}, \boldsymbol{z}_v^t \in \mathbb{R}^{F'}$ with dimensionality $F'$. We denote the input and output representations of $v$, packed together across time, by matrices $\boldsymbol{X}_v \in \mathbb{R}^{T \times D'}$ and $\boldsymbol{Z}_v \in \mathbb{R}^{T \times F'}$ respectively.

The key objective of the temporal self-attentional layer is to capture the temporal variations in graph structure over multiple time steps. The input representation of node $v$ at time-step $t$, $\boldsymbol{x}_v^t$, constitutes an encoding of the current local structure around $v$. We use $\boldsymbol{x}_v^t$ as the query to attend over its historical representations ($< t$), tracing the evolution of the local neighborhood around $v$. Thus, temporal self-attention facilitates learning of dependencies between various representations of a node across different time steps.

To compute the output representation of node $v$ at $t$, we use the scaled dot-product form of attention (Vaswani et al., 2017) where the queries, keys, and values are set as the input node representations. The queries, keys, and values are first transformed to a different space by using linear projections matrices $\boldsymbol{W}_q \in \mathbb{R}^{D' \times F'}, \boldsymbol{W}_k \in \mathbb{R}^{D' \times F'}$ and $\boldsymbol{W}_v \in \mathbb{R}^{D' \times F'}$ respectively. Here, we allow each time-step $t$ to attend over all time-steps up to and including $t$, to prevent leftward information flow and preserve the auto-regressive property. The temporal self-attention is defined as:

$$\boldsymbol{Z}_v = \boldsymbol{\beta_v}(\boldsymbol{X}_v \boldsymbol{W}_v), \quad \beta_v^{ij} = \frac{\exp(e_v^{ij})}{\sum\limits_{k=1}^{T} \exp(e_v^{ik})}, \quad e_v^{ij} = \left( \frac{((\boldsymbol{X}_v \boldsymbol{W}_q)(\boldsymbol{X}_v \boldsymbol{W}_k)^T)_{ij}}{\sqrt{F'}} + M_{ij} \right) \quad (2)$$

where $\boldsymbol{\beta_v} \in \mathbb{R}^{T \times T}$ is the attention weight matrix obtained by the multiplicative attention function and $\boldsymbol{M} \in \mathbb{R}^{T \times T}$ is a mask matrix with each entry $M_{ij} \in \{-\infty, 0\}$. When $M_{ij} = -\infty$, the softmax function results in a zero attention weight, $i.e.$, $\beta_v^{ij} = 0$, which switches off the attention from time-step $i$ to $j$. To encode the temporal order, we define $\boldsymbol{M}$ as:

$$M_{ij} = \begin{cases} 0, & i \leq j \\ -\infty, & \text{otherwise} \end{cases}$$

### 4.3 MULTI-HEAD ATTENTION

We additionally employ multi-head attention (Vaswani et al., 2017) to jointly attend to different subspaces at each input, leading to a leap in model capacity. We use multiple attention heads, followed by concatenation, in both structural and temporal self-attention layers:

$$\text{Structural multi-head self-attention:} \qquad \boldsymbol{h}_v = \text{Concat}(\boldsymbol{z}_v^1, \boldsymbol{z}_v^2, \ldots, \boldsymbol{z}_v^H) \qquad \forall v \in V \qquad (3)$$

$$\text{Temporal multi-head self-attention:} \qquad \boldsymbol{H}_v = \text{Concat}(\boldsymbol{Z}_v^1, \boldsymbol{Z}_v^2, \ldots, \boldsymbol{Z}_v^H) \qquad \forall v \in V \qquad (4)$$

where $H$ is the number of attention heads, $\boldsymbol{h}_v \in \mathbb{R}^F$ and $\boldsymbol{H}_v \in \mathbb{R}^{T \times F'}$ are the outputs of structural and temporal multi-head attentions respectively. Note that while structural attention is applied on a single snapshot, temporal attention operates over multiple time-steps.

### 4.4 DYSAT ARCHITECTURE

In this section, we present our neural architecture DySAT for Dynamic Graph Representation Learning, that uses the above defined *structural* and *temporal* self-attention layers as fundamental modules. As shown in Figure 1, DySAT has three modules from its top to bottom, (1) *structural* attention block, (2) *temporal* attention block, and (3) graph context prediction. The model takes as input a collection of $T$ graph snapshots, and generates outputs latent node representations at each time step.

**Structural attention block.** This module is composed of multiple stacked structural self-attention layers to extract features from nodes at different distances. We apply each layer independently at different snapshots with shared parameters, as illustrated in Figure 1, to capture local neighborhood structure around a node at each time step. Note that the embeddings input to a layer can potentially vary across different snapshots. We denote the node representations output by the structural attention block, as $\{\boldsymbol{h}_v^1, \boldsymbol{h}_v^2, \ldots, \boldsymbol{h}_v^T\}, \boldsymbol{h}_v^t \in \mathbb{R}^f$, which feed as input to the *temporal* attention block.

**Temporal attention block.** First, we equip the temporal attention module with a sense of ordering through *position* embeddings (Gehring et al., 2017), $\{\boldsymbol{p}^1, \ldots, \boldsymbol{p}^T\}, \boldsymbol{p}^t \in \mathbb{R}^f$, which embed the absolute temporal position of each snapshot. The position embeddings are then combined with the output of the structural attention block to obtain a sequence of input representations: $\{\boldsymbol{h}_v^1 + \boldsymbol{p}^1, \boldsymbol{h}_v^2 + \boldsymbol{p}^2, \ldots, \boldsymbol{h}_v^T + \boldsymbol{p}^T\}$ for node $v$ across multiple time steps. This block also follows a similar structure with multiple stacked temporal self-attention layers. The outputs of the final layer pass into a position-wise *feed-forward* layer to give the final node representations $\{\boldsymbol{e}_v^1, \boldsymbol{e}_v^2, \ldots, \boldsymbol{e}_v^T\} \ \forall v \in V$.

**Graph context prediction.** To ensure that the learned representations capture both structural and temporal information, we define an objective function that preserves the local structure around a node, across multiple time steps. We use the dynamic representations of a node $v$ at time step $t$, $\boldsymbol{e}_v^t$ to predict the occurrence of nodes appearing the local neighborhood around $v$ at $t$. In particular, we use a binary cross-entropy loss function at each time step to encourage nodes co-occurring in fixed-length random walks, to have similar representations.

$$L_v = \sum_{t=1}^{T} \sum_{u \in \mathcal{N}_{walk}^t(v)} -\log(\sigma(< \boldsymbol{e}_u^t, \boldsymbol{e}_v^t >)) - w_n \cdot \sum_{u' \in P_n^t(v)} \log(1 - \sigma(< \boldsymbol{e}_{u'}^t, \boldsymbol{e}_v^t >)) \qquad (5)$$

where $\sigma$ is the sigmoid function, $< . >$ denotes the inner product operation, $\mathcal{N}_{walk}^t(v)$ is the set of nodes that co-occur with $v$ on fixed-length random walks at snapshot $t$, $P_n^t$ is a negative sampling distribution for snapshot $\mathcal{G}^t$, and $w_n$, negative sampling ratio, is a tunable hyper-parameter to balance the positive and negative samples.

## 5 EXPERIMENTS

We evaluate the quality of our learned node representations on the fundamental task of dynamic link prediction. We choose this task since it has been widely used (Trivedi et al., 2017; Goyal et al., 2017; Li et al., 2018) in evaluating the quality of dynamic node representations to predict the temporal evolution in graph structure.

In our experiments, we compare the performance of DySAT against a variety of static and dynamic graph representation learning baselines. Our experimental results on four publicly available benchmarks indicate that DySAT achieves significant performance gains over other methods.

| Dataset | Communication Networks | | Rating Networks | |
|---|---|---|---|---|
| | Enron | UCI | Yelp | ML-10M |
| # Nodes | 143 | 1,809 | 6,569 | 20,537 |
| # Links | 2,347 | 16,822 | 95,361 | 43,760 |
| # Time steps | 10 | 13 | 12 | 13 |

Table 1: Statistics of the datasets used in our experiments

## 5.1 DATASETS

We use four dynamic graph datasets with two communication and bipartite rating networks each.

**Communication networks.** We consider two publicly available communication network datasets: Enron (Klimt & Yang, 2004) and UCI (Panzarasa et al., 2009). In Enron, the communication links are email interactions between core employees and the links in UCI represent messages sent between users on an online social network platform.

**Rating networks.** We use two bipartite rating networks from Yelp[1] and MovieLens (Harper & Konstan, 2016). In Yelp, the dynamic graph comprises links between two types of nodes, users and businesses, derived from the observed ratings over time. ML-10M consists of a user-tag interaction network where user-tag links connects users with the tags they applied on certain movies.

In each dataset, multiple graph snapshots are created based on the observed interactions in fixed-length time windows. Dataset statistics are shown in Table 1, while Appendix G has further details.

## 5.2 EXPERIMENTAL SETUP

We conduct experiments on the task of link prediction in dynamic graphs, where we learn dynamic node representations on snapshots $\{\mathcal{G}^1, \ldots, \mathcal{G}^t\}$ and use $\{e_v^t, \forall v \in \mathcal{V}\}$ to predict the links at $\mathcal{G}^{t+1}$ during evaluation. We compare different models based on their ability to correctly classify each example (node pair) into links and non-links. To further analyze predictive capability, we also evaluate *new* link prediction, with a focus on new links that appear at each time step, (Appendix B).

We evaluate the performance of different models by training a logistic regression classifier for dynamic link prediction (Zhou et al., 2018). We create evaluation examples from the links in $\mathcal{G}^{t+1}$ and an equal number of randomly sampled pairs of unconnected nodes (non-links). A held-out validation set (20% links) is used to tune the hyper-parameters across all models, which is later discarded. We randomly sample 25% of the examples for training and use the remaining 75% as our test set. We repeat this for 10 randomized runs and report the average performance in our results.

We follow the strategy recommended by Grover & Leskovec (2016) to compute the feature representation for a pair of nodes, using the Hadamard Operator ($e_u^t \odot e_v^t$), for all methods unless explicitly specified otherwise. The Hadamard operator computes the element-wise product of two vectors and closely mirrors the widely used inner product operation in learning node embeddings. We evaluate the performance of link prediction using Area Under the ROC Curve (AUC) scores (Grover & Leskovec, 2016). We also report the average precision scores in Table 6 of the Appendix.

We implement DySAT in Tensorflow (Abadi et al., 2016) and employ mini-batch gradient descent with Adam optimizer (Kingma & Ba, 2015) for training. For Enron, we use a single layer in both the structural and temporal blocks, with each layer comprising 16 attention heads computing 8 features apiece (for a total of 128 dimensions). In the other datasets, we use two structural self-attentional layers with 16 and 8 heads respectively, each computing 16 features (layer sizes of 256, 128). The model is trained for a maximum of 200 epochs with a batch size of 256 nodes and the best performing model on the validation set, is chosen for evaluation.

## 5.3 BASELINE

We compare the performance of DySAT with several state-of-the-art dynamic graph embedding techniques. In addition, we include several static graph embedding methods in comparison to ana-

---

[1]https://www.yelp.com/dataset/challenge

| Method | Enron | | UCI | | Yelp | | ML-10M | |
|---|---|---|---|---|---|---|---|---|
| | Micro-AUC | Macro-AUC | Micro-AUC | Macro-AUC | Micro-AUC | Macro-AUC | Micro-AUC | Macro-AUC |
| node2vec | $83.72 \pm 0.7$ | $83.05 \pm 1.2$ | $79.99 \pm 0.4$ | $80.49 \pm 0.6$ | $67.86 \pm 0.2$ | $65.34 \pm 0.2$ | $87.74 \pm 0.2$ | $87.52 \pm 0.3$ |
| G-SAGE | $82.48^* \pm 0.6$ | $81.88^* \pm 0.5$ | $79.15^* \pm 0.4$ | $82.89^* \pm 0.2$ | $60.95^\dagger \pm 0.1$ | $58.56^\dagger \pm 0.2$ | $86.19^\ddagger \pm 0.3$ | $89.92^\ddagger \pm 0.1$ |
| G-SAGE + GAT | $72.52 \pm 0.4$ | $73.34 \pm 0.6$ | $74.03 \pm 0.4$ | $79.83 \pm 0.2$ | $66.15 \pm 0.1$ | $65.09 \pm 0.2$ | $83.97 \pm 0.3$ | $84.93 \pm 0.1$ |
| GCN-AE | $81.55 \pm 1.5$ | $81.71 \pm 1.5$ | $80.53 \pm 0.3$ | $83.50 \pm 0.5$ | $66.71 \pm 0.2$ | $65.82 \pm 0.2$ | $85.49 \pm 0.1$ | $85.74 \pm 0.1$ |
| GAT-AE | $75.71 \pm 1.1$ | $75.97 \pm 1.4$ | $79.98 \pm 0.2$ | $81.86 \pm 0.3$ | $65.92 \pm 0.1$ | $65.37 \pm 0.1$ | $87.01 \pm 0.2$ | $86.75 \pm 0.2$ |
| DynamicTriad | $80.26 \pm 0.8$ | $78.98 \pm 0.9$ | $77.59 \pm 0.6$ | $80.28 \pm 0.5$ | $63.53 \pm 0.3$ | $62.69 \pm 0.3$ | $88.71 \pm 0.2$ | $88.43 \pm 0.1$ |
| Know-Evolve | $61.57 \pm 1.1$ | $62.28 \pm 1.5$ | $71.20 \pm 0.5$ | $80.93 \pm 0.2$ | $56.88 \pm 0.2$ | $59.68 \pm 0.2$ | $78.80 \pm 0.5$ | $83.70 \pm 0.2$ |
| DynGEM | $67.83 \pm 0.6$ | $69.72 \pm 1.3$ | $77.49 \pm 0.3$ | $79.82 \pm 0.5$ | $66.02 \pm 0.2$ | $65.94 \pm 0.2$ | $73.69 \pm 1.2$ | $85.96 \pm 0.3$ |
| **DySAT** | $\mathbf{85.71} \pm 0.3$ | $\mathbf{86.60} \pm 0.2$ | $\mathbf{81.03} \pm 0.2$ | $\mathbf{85.81} \pm 0.1$ | $\mathbf{70.15} \pm 0.1$ | $\mathbf{69.87} \pm 0.1$ | $\mathbf{90.82} \pm 0.3$ | $\mathbf{93.68} \pm 0.1$ |

Table 2: Experiment results on dynamic link prediction (micro and macro averaged AUC with standard deviation). We show GraphSAGE (denoted by G-SAGE) results with the best performing aggregators for each dataset ($*$ represents GCN, $\dagger$ represents LSTM, and $\ddagger$ represents max-pooling).

lyze the gains of using temporal information for dynamic link prediction. To make a fair comparison with static methods, we provide access to the entire history of snapshots by constructing an aggregated graph upto time $t$, where the weight of each link is defined as the cumulative weight till $t$ agnostic of its occurrence times. We use author-provided implementations for all the baselines and set the final embedding dimension $d = 128$.

We compare against several state-of-the-art unsupervised static embedding methods: node2vec (Grover & Leskovec, 2016), GraphSAGE (Hamilton et al., 2017b) and graph autoencoders (Hamilton et al., 2017a). We experiment with different aggregators in GraphSAGE, namely, GCN, mean-pooling, max-pooling, and LSTM, to report the performance of the best performing aggregator in each dataset. To provide a fair comparison with GAT (Velickovic et al., 2018), which originally conduct experiments only on node classification, we implement a graph attention layer as an additional aggregator in GraphSAGE, which we denote by GraphSAGE + GAT. We also train GCN and GAT as autoencoders for link prediction along the suggested lines of (Zitnik et al., 2018), denoted by GCN-AE and GAT-AE respectively. In the dynamic setting, we evaluate DySAT against the most recent studies on dynamic graph embedding including Know-Evolve (Trivedi et al., 2017), DynamicTriad (Zhou et al., 2018), and DynGEM (Goyal et al., 2017). The details of hyper-parameter tuning for all methods can be found in Appendix F.

## 5.4 Experimental Results

We evaluate the models at each time step $t$ by training separate models up to snapshot $t$ and evaluate at $t + 1$ for each $t = 1, \ldots, T$. We summarize the micro and macro averaged AUC scores (across all time steps) for all models in Table 2. From the results, we observe that DySAT achieves consistent gains of 3–4% macro-AUC, in comparison to the best baseline across all datasets.

Further, we compare the model performance at each time step (Figure 2), to obtain a deep understanding of their temporal behaviors. We fine the performance of DySAT to be relatively more stable than other methods. This contrast is pronounced in the communication networks (Enron and UCI), where we observe drastic drops in performance of static embedding methods at certain time steps.

The runtime per mini-batch of DySAT on ML-10M, using a machine with Nvidia Tesla V100 GPU and 28 CPU cores, is 0.72 seconds. In comparison, a model variant without temporal attention (Appendix A) takes 0.51 seconds, which illustrates the relatively low cost of temporal attention.

## 6 Discussion

Our experimental results provide several interesting observations and insights to the performance of different graph embedding techniques.

First, we observe that GraphSAGE achieves comparable performance to DynamicTriad across different datasets, despite being trained only on static graphs. One possible explanation may be that GraphSAGE uses trainable neighbor-aggregation functions, while DynamicTriad employs Skip-gram based methods augmented with temporal smoothness. This leads us to conjecture that the combination of structural and temporal modeling with expressive aggregation functions, such as

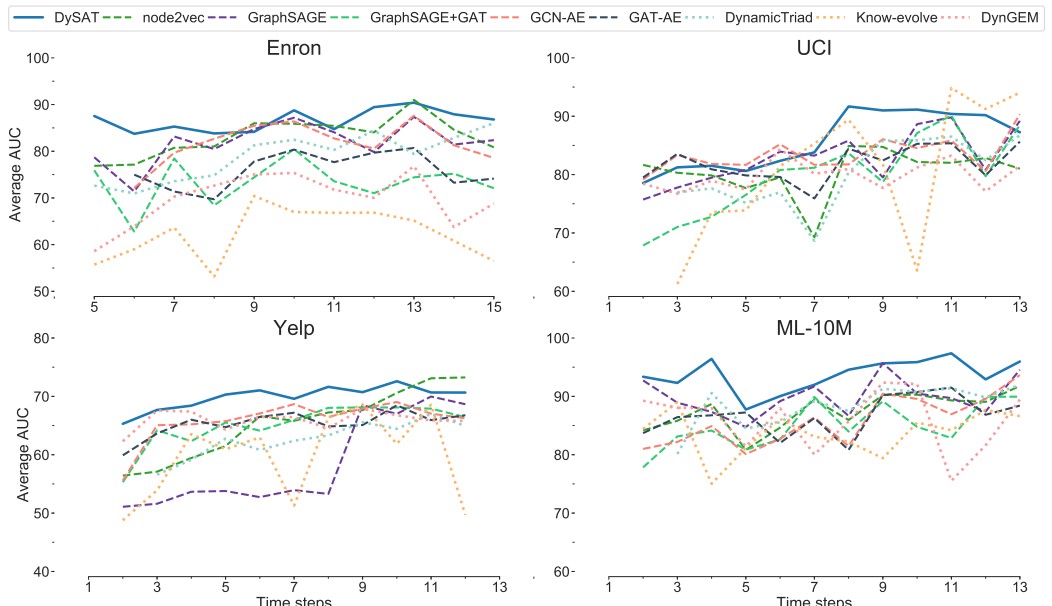

Figure 2: Performance comparison of DySAT with different models across multiple time steps: the solid line represents DySAT; dashed lines represent static graph embedding models; and dotted lines represent dynamic graph embedding models. We truncate the y-axis to avoid visual clutter.

multi-head attention, is responsible for the consistently superior performance of DySAT on dynamic link prediction. We also observe that node2vec achieves consistent performance agnostic of temporal information, which demonstrates the effectiveness of second-order random walk sampling. This observation points to the direction of applying sampling techniques to further improve DySAT.

In DySAT, we employ structural attention layers followed by temporal attention layers. We choose this design because graph structures are not stable over time, which makes directly employing structural attention layers after temporal attention layers infeasible. We also consider another alternative design choice that applies self-attention along the two dimensions of neighbors and time together following the strategy similar to (Shen et al., 2018). In practice, this would be computationally expensive due to variable number of neighbors per node across multiple snapshots. We leave exploring other architectural design choices based on structural and temporal self-attentions as future work.

In the current setup, we store the adjacency matrix of each snapshot in memory using sparse matrix, which may pose memory challenges when scaling to large graphs. In the future, we plan to explore DySAT with memory-efficient mini-batch training strategy along the lines of Graph-SAGE (Hamilton et al., 2017b). Further, we develop an incremental self-attention network (IncSAT) that is efficient in both computation and memory cost as a direct extension of DySAT. Our initial results are promising as reported in Appendix E, which opens the door to future exploration of self-attentional architectures for incremental (or streaming) graph representation learning. We also evaluate the capability of DySAT on multi-step link prediction or forecasting and observe significant relative improvements of 6% AUC on average over existing methods, as reported in Appendix C.

## 7 CONCLUSION

In this paper, we introduce a novel self-attentional neural network architecture named DySAT to learn node representations in dynamic graphs. Specifically, DySAT computes dynamic node representations using self-attention over the (1) structural neighborhood and (2) historical node representations, thus effectively captures the temporal evolutionary patterns of graph structures. Our experiment results on various real-world dynamic graph datasets indicate that DySAT achieves significant performance gains over several state-of-the-art static and dynamic graph embedding baselines. Though our experiments are conducted on graphs without node features, DySAT can be easily generalized on feature-rich graphs. Another interesting direction is exploring continuous-time generalization of our framework to incorporate more fine-grained temporal variations.

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

# A ANALYSIS OF TEMPORAL ATTENTION

In this section, we conduct an in-depth analysis of the proposed temporal attention layer, to demonstrate its utility and examine the distribution of temporal attention weights.

## A.1 EFFECT OF REMOVING TEMPORAL LAYERS

To demonstrate the effectiveness of temporal self-attention layers, we conduct an experimental study that removes the temporal attention block from DySAT to create a simpler architecture. This model is optimized using the same loss function (Eqn. 5) applied on the intermediate representations $\{h_v^1, h_v^2, \ldots, h_v^T\}$ for each node $v \in \mathcal{V}$. Note that this model is different from static methods since the structural self-attention block is jointly optimized (using Eqn. 5) across all snapshots without any explicit temporal modeling. We use the best configuration of DySAT on each dataset from the original experiments to initialize the new model. The performance comparison is shown in Table 3. We observe that in some datasets, the structural attention block is able to learn some temporal evolution patterns in graph structure, despite the lack of explicit temporal modeling. However, the new model is consistently inferior to DySAT and we observe that the original DySAT has a 3% average gain in Macro-AUC, which validates our choice of using temporal self-attentional layers.

| Method | Enron | | UCI | | Yelp | | ML-10M | |
|---|---|---|---|---|---|---|---|---|
| | Micro-AUC | Macro-AUC | Micro-AUC | Macro-AUC | Micro-AUC | Macro-AUC | Micro-AUC | Macro-AUC |
| Original | **85.71** ± 0.3 | **86.60** ± 0.2 | **81.03** ± 0.2 | **85.81** ± 0.1 | **70.15** ± 0.1 | **69.87** ± 0.1 | **90.82** ± 0.3 | **93.68** ± 0.1 |
| No Temporal | 84.50 ± 0.3 | 85.68 ± 0.4 | 76.61 ± 0.2 | 79.97 ± 0.3 | 68.34 ± 0.1 | 67.20 ± 0.3 | 89.61 ± 0.4 | 91.10 ± 0.2 |

Table 3: Experimental study on removing temporal attention layers from DySAT (micro and macro averaged AUC with standard deviation)

## A.2 VISUALIZATION OF TEMPORAL ATTENTION WEIGHTS

We conduct a qualitative analysis to obtain deeper insights into the distribution of temporal attention weights learned by DySAT. In this experiment, we examine the temporal attention coefficients learned at each time step $t$, which indicate the relative importance of each historical snapshot ($< t$) in predicting the links at $t$. We choose the Enron dataset to visualize the mean and standard deviation of temporal attention coefficients, over all the nodes. Figure 3 visualizes a heatmap of the learned temporal attention weights on Enron dataset for the first 10 time steps.

From Figure 3, we observe that the mean temporal attention weights are mildly biased towards recent snapshots, while the historical snaphots vary in their importance across different time steps. Further, we find that the standard deviation of attention weights across different nodes is generally high and exhibits more variability. Thus, the temporal attention weights are well distributed across historical snapshots, with significant variance across different nodes in the graph. While this analysis attempts to provide a high-level perspective on the weights learned by temporal attention, an appropriate interpretation of these coefficients (as done by *e.g.*, (Bahdanau et al., 2015)) requires further domain knowlege about the dataset under study, and is left as future work.

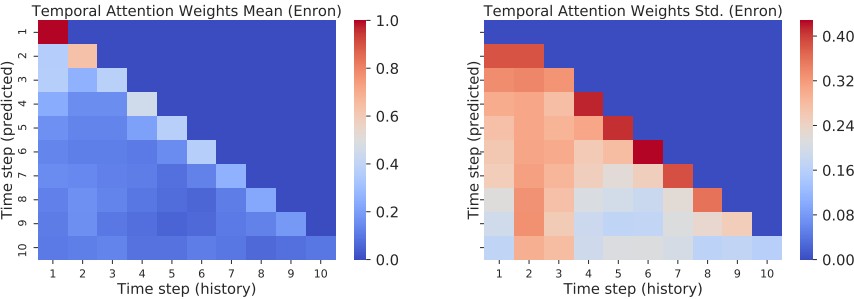

Figure 3: Heatmap visualizing mean and standard deviation of temporal attention weights over all nodes in Enron dataset.

# B    DYNAMIC NEW LINK PREDICTION

In this section, we additionally report the results of dynamic link prediction evaluated only on the *new* links at each time step. This provides an in-depth analysis on the capabilities of different methods in predicting relatively unseen links. We follow the same evaluation setup of training a downstream logistic regression classifier for dynamic link prediction. However, a key difference is that the evaluation examples comprise new links at $\mathcal{G}_{t+1}$ (that are not in $\mathcal{G}_t$) and an equal number of randomly sampled non-links.

Table 4 summarizes the micro and macro averaged AUC scores for different methods on the four datasets. The absolute performance numbers of all methods are lower than the original evaluation setup of using all links at $\mathcal{G}_{t+1}$, which is reasonable since accurate prediction of new links at $\mathcal{G}_{t+1}$ is expected to be slightly more challenging in comparison to predicting all the links at $\mathcal{G}_{t+1}$. From Table 2), we find that DySAT achieves consistent relative gains of 3–5% Macro-AUC over the best baselines on dynamic new link prediction as well, thus validating its effectiveness in accurately capturing temporal context for new link prediction.

| Method | Enron | | UCI | | Yelp | | ML-10M | |
|---|---|---|---|---|---|---|---|---|
| | **Micro-AUC** | **Macro-AUC** | **Micro-AUC** | **Macro-AUC** | **Micro-AUC** | **Macro-AUC** | **Micro-AUC** | **Macro-AUC** |
| node2vec | $76.92 \pm 1.2$ | $75.86 \pm 0.5$ | $73.67 \pm 0.3$ | $74.76 \pm 0.8$ | $67.36 \pm 0.2$ | $65.17 \pm 0.2$ | $85.22 \pm 0.2$ | $84.89 \pm 0.1$ |
| G-SAGE | $73.92^* \pm 0.7$ | $74.67^* \pm 0.6$ | $76.69^* \pm 0.3$ | $79.41^* \pm 0.1$ | $62.25^\dagger \pm 0.2$ | $58.81^\dagger \pm 0.3$ | $85.23^\ddagger \pm 0.3$ | $89.14^\ddagger \pm 0.2$ |
| G-SAGE + GAT | $67.02 \pm 0.8$ | $68.32 \pm 0.7$ | $73.18 \pm 0.4$ | $76.79 \pm 0.2$ | $66.53 \pm 0.2$ | $65.45 \pm 0.1$ | $80.84 \pm 0.3$ | $82.53 \pm 0.1$ |
| GCN-AE | $74.46 \pm 1.1$ | $74.02 \pm 1.6$ | $74.76 \pm 0.1$ | $76.75 \pm 0.6$ | $66.18 \pm 0.2$ | $65.77 \pm 0.3$ | $82.45 \pm 0.3$ | $82.48 \pm 0.2$ |
| GAT-AE | $69.75 \pm 2.2$ | $69.25 \pm 1.9$ | $72.52 \pm 0.4$ | $73.78 \pm 0.7$ | $66.07 \pm 0.1$ | $65.91 \pm 0.2$ | $84.98 \pm 0.2$ | $84.51 \pm 0.3$ |
| DynamicTriad | $69.59 \pm 1.2$ | $68.77 \pm 1.7$ | $67.97 \pm 0.7$ | $71.67 \pm 0.9$ | $63.76 \pm 0.2$ | $62.83 \pm 0.3$ | $84.72 \pm 0.2$ | $84.32 \pm 0.2$ |
| Know-Evolve | $59.05 \pm 2.7$ | $59.63 \pm 2.7$ | $69.10 \pm 0.3$ | $77.48 \pm 0.2$ | $56.95 \pm 0.2$ | $59.72 \pm 0.5$ | $76.83 \pm 0.5$ | $82.23 \pm 0.2$ |
| DynGEM | $60.73 \pm 1.1$ | $62.85 \pm 1.9$ | $77.49 \pm 0.3$ | $79.82 \pm 0.5$ | $66.42 \pm 0.2$ | $66.84 \pm 0.2$ | $73.77 \pm 0.7$ | $83.51 \pm 0.3$ |
| **DySAT** | $\mathbf{78.87} \pm 0.6$ | $\mathbf{78.58} \pm 0.6$ | $\mathbf{79.24} \pm 0.3$ | $\mathbf{83.66} \pm 0.2$ | $\mathbf{69.46} \pm 0.1$ | $\mathbf{69.14} \pm 0.1$ | $\mathbf{89.29} \pm 0.2$ | $\mathbf{92.65} \pm 0.1$ |

Table 4: Experiment results on dynamic *new* link prediction (micro and macro averaged AUC with standard deviation). We show GraphSAGE (denoted by G-SAGE) results with the best performing aggregators for each dataset ($*$ represents GCN, $\dagger$ represents LSTM, and $\ddagger$ represents max-pooling).

# C    MULTI-STEP LINK PREDICTION

In this section, we evaluate various dynamic graph representation learning methods on the task of multi-step link prediction or forecasting. Here, each model is trained for a fixed number of time steps, and the latest embeddings are used to predict links at multiple future time steps. In each dataset, we choose the last 6 snapshots to evaluate multi-step link prediction. The model is trained on the previous remaining snapshots, and the latest embeddings are used to forecast links at future time steps. For each future time step $t + \Delta$ $(1 \leq \Delta \leq 6)$, we create examples from the links in $\mathcal{G}_{t+\Delta}$ and an equal number of randomly sampled pairs of unconnected nodes (non-links). We otherwise use the same evaluation setup of training a downstream logistic regression classifier to evaluate link prediction. In this experiment, we exclude the links formed by nodes that newly appear in the future evaluation snapshots, since most methods cannot be easily support updates for new nodes.

Figure. 4 depicts the variation in model performance of different methods over the 6 evaluation snapshots. As expected, we observe a slight decay in performance over time for all the models. DySATachieves significant performance gains over all other baselines and maintains a highly stable link prediction performance over multiple future time steps. Static embedding methods often exhibit large variations in performance over time steps, while DySAT achieves a stable consistent performance. The historical context captured by the dynamic node embeddings of DySAT, is one of the most likely reasons for its stable multi-step forecasting performance.

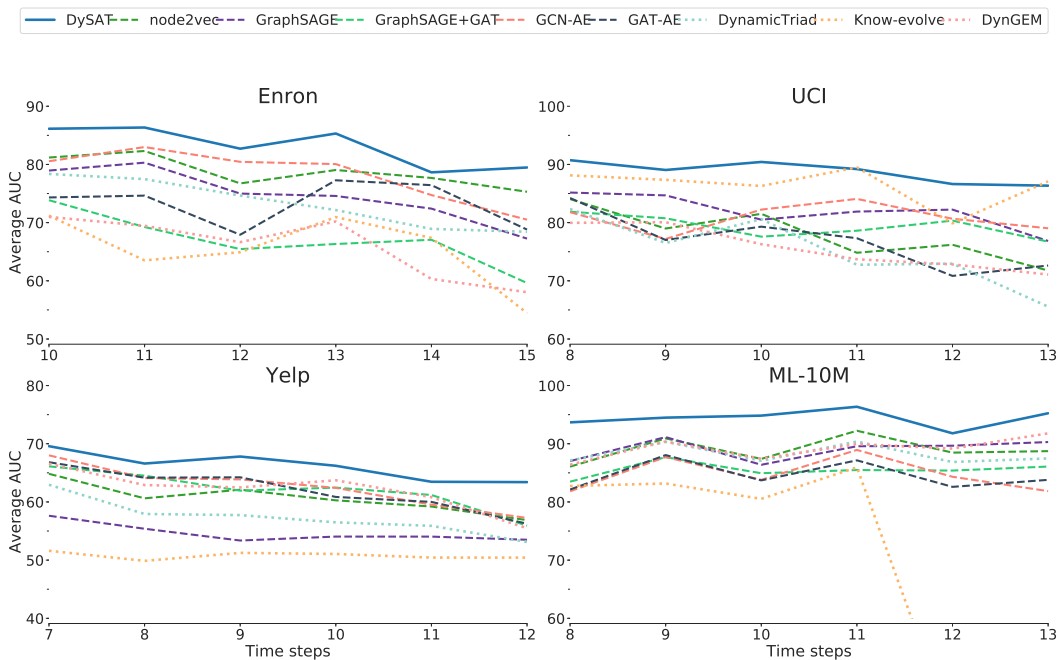

Figure 4: Performance comparison of DySAT with different models on multi-step link prediction for 6 future time steps on all datasets

# D IMPACT OF UNSEEN NODES ON DYNAMIC LINK PREDICTION

In this section, we analyze the sensitivity of different graph representation learning on link prediction for previously unseen nodes that appear newly at time $t$.

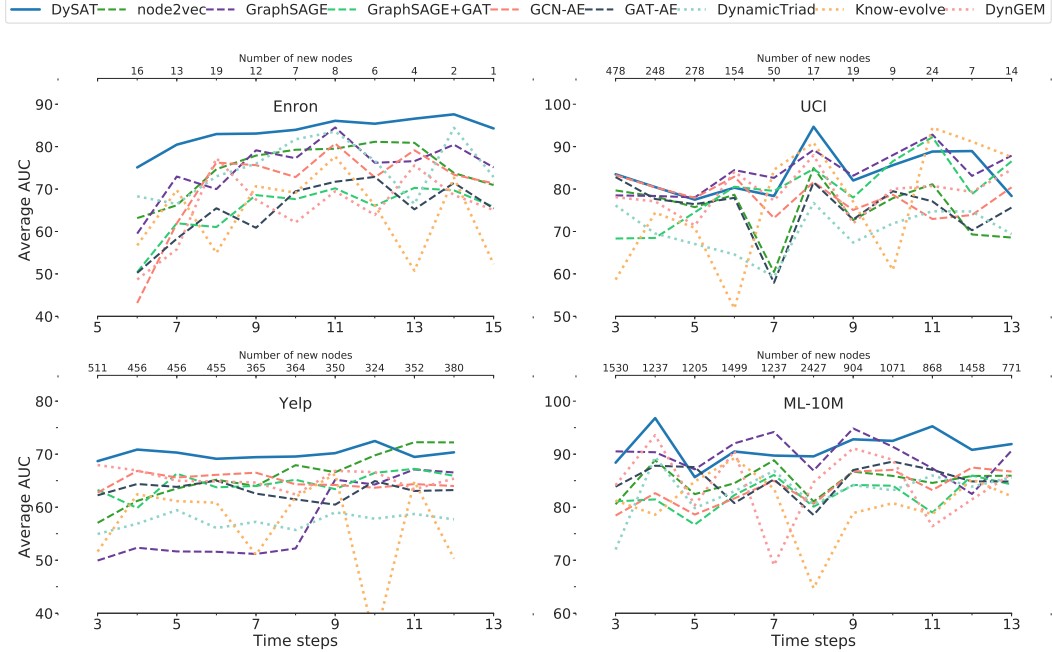

Figure 5: Performance comparison of DySAT with different models on link prediction restricted to *new* nodes at each time step

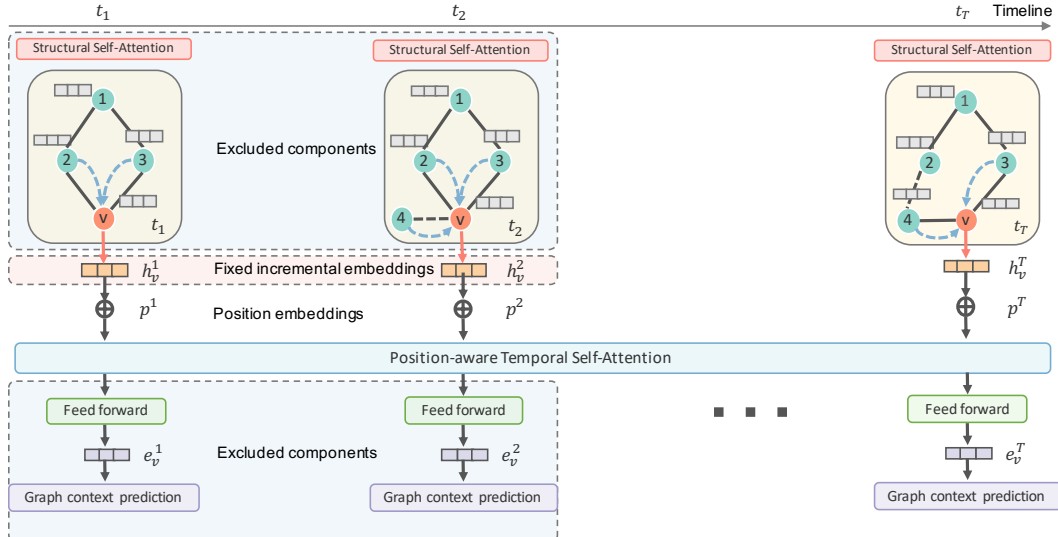

Figure 6: Neural architecture of IncSAT: the components that are excluded from DySAT are wrapped by dashed blue rectangles. The intermediate node representations are directly loaded from saved models trained previously.

A node is considered as a *new* node at time step $t$ in $\mathcal{G}_t$ if it has not appeared (has no links) in any of the previous $t - 1$ snapshots. In this experiment, the evaluation set at time step $t$ only comprises the subset of links at $\mathcal{G}_{t+1}$ among the new nodes in $\mathcal{G}_t$ and corresponding randomly sampled non-links. Since the number of nodes varies significantly across different time steps, we report the performance of each method along with the number of new nodes at each time step, in Figure 5.

From Figure 5, we observe that DySAT outperforms other baselines in most datasets, demonstrating the ability to characterize new or previously unseen nodes despite their limited history. Although the temporal attention will focus on the latest representation of a new node $v$ due to absence of history, the structural embedding of $v$ recieves backpropagation signals through the temporal attention on neighboring nodes, which indirectly affects the final embedding of $v$. We hypothesize that this indirect temporal signal is one of the reasons for DySAT to achieve performance improvements over baselines, albeit not designed to explicitly model historical context for previously unseen nodes.

# E  INCREMENTAL SELF-ATTENTION NETWORK

In this section, we describe an extension of our dynamic self-attentional architecture to learn incremental node representations. The motivation of incremental learning arises due to the proliferation in sizes of real-world graphs, making it difficult to store multiple snapshots in memory. Thus, the incremental graph representation learning problem imposes the restriction of no access to historical graph snapshots, in contrast to most dynamic graph embedding methods. Specifically, to learn the node embeddings $\{e_v^t \in \mathbb{R}^d \ \forall v \in \mathcal{V}\}$ at time $T$, we require a model to only access to the snapshot $\mathcal{G}^T$ and a *summary* of the historical snapshots. For example, DynGEM (Goyal et al., 2017) is an example of an incremental embedding method that uses the embeddings learned at step $t - 1$, as initialization to learn the embeddings at $t$.

We propose an extension of our self-attentional architecture named IncSAT to explore solving the incremental graph representation learning problem. To learn node representations at $T$, we first incrementally train multiple models at $1 \le t \le T$. Unlike the original DySAT where structural self-attention is applied at each snapshot, IncSAT applies the structural block only at the latest graph snapshot $\mathcal{G}^t$. We enable incremental learning by storing the intermediate output representations $\{h_v^T \ \forall v \in \mathcal{V}\}$ of the structural block. As illustrated in Figure 6, these intermediate output representations of historical snapshots ($1 \le t < T$) can be directly loaded from previously saved results at $1 \le t < T$. Thus, the structural information of the previous historical snapshots are *summarized* in the stored intermediate representations. The temporal self-attention is only applied to the current

| Method | Enron | | UCI | | Yelp | | ML-10M | |
|--------|-------|-------|-------|-------|-------|-------|--------|-------|
| | Micro-AUC | Macro-AUC | Micro-AUC | Macro-AUC | Micro-AUC | Macro-AUC | Micro-AUC | Macro-AUC |
| **DySAT** | **85.71** $\pm$ 0.3 | **86.60** $\pm$ 0.2 | **81.03** $\pm$ 0.2 | **85.81** $\pm$ 0.1 | **70.15** $\pm$ 0.1 | **69.87** $\pm$ 0.1 | **90.82** $\pm$ 0.3 | **93.68** $\pm$ 0.1 |
| DynGEM | 67.83 $\pm$ 0.6 | 69.72 $\pm$ 1.3 | 77.49 $\pm$ 0.3 | 79.82 $\pm$ 0.5 | 66.02 $\pm$ 0.2 | 65.94 $\pm$ 0.2 | 73.69 $\pm$ 1.2 | 85.96 $\pm$ 0.3 |
| IncSAT | 84.36 $\pm$ 0.2 | 85.43 $\pm$ 0.3 | 76.18 $\pm$ 0.5 | 85.37 $\pm$ 0.2 | 69.54 $\pm$ 0.1 | 68.73 $\pm$ 0.3 | 80.13 $\pm$ 0.4 | 91.14 $\pm$ 0.2 |

Table 5: Experimental results of IncSAT in comparison to DySAT (micro and macro averaged AUC with standard deviation)

snapshot $\mathcal{G}^T$ over the historical representations of each node to compute the final node embeddings $\{e_v^T \ \forall v \in \mathcal{V}\}$ at $T$, which are trained on random walks sampled from $\mathcal{G}^T$.

We evaluate IncSAT using the same experimental setup, with minor modifications in hyper-parameters. We use a dropout rate of 0.4 in both the structural and temporal self-attention layers. From our preliminary experiments, we find that a higher dropout rate in the structural block can facilitate avoiding over-fitting the model to the current graph snapshot. In Table 5, we report the performance of IncSAT in comparison to DySAT and DynGEM, which is the only one that can support incremental training from our baseline models. The results show that IncSAT achieves comparable performance to DySAT on most datasets while significantly outperforming DynGEM, albeit with minimal hyper-parameter tuning.

## F  DETAILS ON HYPER-PARAMETER SETTINGS AND TUNING

In DySAT, the objective function (Eqn. 5) utilizes positive pairs of nodes co-occurring in fixed-length random walks. We follow the strategy of Deepwalk (Perozzi et al., 2014) to sample walks 10 walks of length 40 per node, each with a context window size of 10. We use 10 negative samples per positive pair, with context distribution ($P_n^t$) smoothing over node degrees with a smoothing parameter of 0.75, following (Perozzi et al., 2014; Grover & Leskovec, 2016; Hamilton et al., 2017b). During training, we apply $L_2$ regularization with $\lambda = 5 \times 10^{-4}$ and use dropout rates (Srivastava et al., 2014) of 0.1 and 0.5 in the self-attention layers of the structural and temporal blocks respectively. We use the validation set for tuning the learning rate in the range of $\{10^{-4}, 10^{-3}\}$ and negative sampling ratio $w_n$ in the range $\{0.01, 0.1, 1\}$.

We tune the hyper-parameters of all baselines following their recommended guidelines. For node2vec, we use the default settings as in the paper, with 10 random walks of length 80 per node and context window of 10, trained for a single epoch. We tune the in-out and return hyper-parameters, $p, q$ using grid-search, in the range $\{0.25, 0.50, 1, 2, 4\}$ and report the best results. In case of Graph-SAGE, we train a two layer model with respective neighborhood sample sizes 25 and 10, for 10 epochs, as described in the original paper. We evaluate the embeddings at each epoch on the validation set, and choose the best for final evaluation. Note that the results of GraphSAGE reported in Table 2 represent that of best-performing aggregator in each dataset.

For Know-evolve, we tune the two weight-scale hyper-parameters in the range $\{10^{-4}, 10^{-3}, 0.01, 0.1\}$, learning rates in $\{10^{-4}, 10^{-3}\}$ and choose the best performing model. DynamicTriad (Zhou et al., 2018) was tuned using their two key hyper-parameters determining the effect of smoothness and triadic closure, $\beta_0$ and $\beta_1$ in the range $\{0.01, 0.1, 1, 10\}$, as advised, while using recommended settings otherwise. We use the $L_1$ operator ($|e_u^t - e_u^t|$) instead of Hadamard, as recommended in the paper, which also gives better performance. For DynGEM, we tune the different scaling and regularization hyper-parameters, $\alpha \in \{10^{-6}, 10^{-5}\}$, $\beta \in \{0.1, 1, 2, 5\}$, $\nu_1 \in \{10^{-6}, 10^{-4}\}$ and $\nu_2 \in \{10^{-6}, 10^{-4}\}$, while using other default configurations.

## G  ADDITIONAL DATASET DETAILS

In this section, we provide some additional, relevant dataset details. Since dynamic graphs often contain continuous timestamps, we split the data into multiple snapshots using suitable time-windows such each snapshot has an equitable yet reasonable number of interactions (communication/ratings). In each snapshot, the weight of a link is determined by the number of interactions between the cor-

| Method | Enron | | UCI | | Yelp | | ML-10M | |
|---|---|---|---|---|---|---|---|---|
| | Micro-AP | Macro-AP | Micro-AP | Macro-AP | Micro-AP | Macro-AP | Micro-AP | Macro-AP |
| node2vec | $84.26 \pm 0.8$ | $84.11 \pm 1.1$ | $80.22 \pm 0.4$ | $81.12 \pm 0.5$ | $66.46 \pm 0.2$ | $63.82 \pm 0.2$ | $88.86 \pm 0.2$ | $88.71 \pm 0.3$ |
| G-SAGE | $83.99^* \pm 0.6$ | $84.02^* \pm 0.6$ | $75.91^* \pm 0.6$ | $82.36^* \pm 0.2$ | $58.81^\dagger \pm 0.1$ | $55.84^\dagger \pm 0.2$ | $85.45^\ddagger \pm 0.3$ | $90.26^\ddagger \pm 0.2$ |
| G-SAGE + GAT | $72.60 \pm 0.5$ | $74.75 \pm 0.9$ | $66.77 \pm 0.4$ | $76.30 \pm 0.4$ | $62.43 \pm 0.1$ | $61.49 \pm 0.3$ | $81.69 \pm 0.6$ | $82.35 \pm 0.2$ |
| GCN-AE | $81.97 \pm 1.5$ | $83.08 \pm 1.4$ | $80.73 \pm 0.4$ | $84.16 \pm 0.6$ | $65.92 \pm 0.2$ | $65.39 \pm 0.2$ | $86.85 \pm 0.2$ | $87.43 \pm 0.1$ |
| GAT-AE | $76.98 \pm 1.1$ | $78.18 \pm 0.8$ | $80.14 \pm 0.4$ | $83.75 \pm 0.3$ | $65.45 \pm 0.2$ | $65.01 \pm 0.2$ | $88.42 \pm 0.2$ | $88.14 \pm 0.2$ |
| DynamicTriad | $82.06 \pm 0.9$ | $81.22 \pm 0.9$ | $76.21 \pm 0.8$ | $80.05 \pm 0.6$ | $61.29 \pm 0.3$ | $60.79 \pm 0.3$ | $89.91 \pm 0.2$ | $89.61 \pm 0.3$ |
| Know-Evolve | $57.68 \pm 1.2$ | $60.71 \pm 1.7$ | $66.99 \pm 0.5$ | $77.49 \pm 0.2$ | $53.98 \pm 0.2$ | $56.44 \pm 0.2$ | $75.64 \pm 0.5$ | $79.97 \pm 0.3$ |
| DynGEM | $70.37 \pm 0.5$ | $72.35 \pm 1.0$ | $78.78 \pm 0.3$ | $81.71 \pm 0.3$ | $\mathbf{68.02} \pm 0.2$ | $\mathbf{68.09} \pm 0.2$ | $80.65 \pm 0.9$ | $89.43 \pm 0.2$ |
| DySAT | $\mathbf{86.82} \pm 0.3$ | $\mathbf{88.25} \pm 0.2$ | $\mathbf{80.88} \pm 0.2$ | $\mathbf{85.96} \pm 0.1$ | $65.81 \pm 0.1$ | $66.76 \pm 0.1$ | $\mathbf{93.03} \pm 0.2$ | $\mathbf{94.92} \pm 0.1$ |

Table 6: Experiment results on dynamic link prediction (micro and macro average precision with standard deviation). We show GraphSAGE (denoted by G-SAGE) results with the base performing aggregators for each dataset (∗ represents GCN, † represents LSTM, and ‡ represents max-pooling).

responding pair of users during that time-period. The pre-processed versions of all datasets will be made publicly available, along with the scripts used for processing the raw data.

**Communication Networks.** The original un-processed Enron dataset is available at `https://www.cs.cmu.edu/~./enron/`. We use only the email communcations that are between Enron employees, *i.e.*, sent by an Enron employee and have at least one recipient who is an Enron employee. A time-window of 2 months is used to construct 16 snapshots, where the first 5 are used as warm-up (due to sparsity) and the remaining 11 snapshots for evaluation.

The UCI dataset was downloaded from `http://networkrepository.com/opsahl_ucsocial.php`. This dataset contains private messages sent between users over a span of six months, on an online social network platform at the University of California, Irvine. The snapshots are created using their communication history with a time-window of 10 days. We discard/merge the terminal snapshots if they do not contain sufficient communications.

**Rating Networks.** We use the Round 11 version of the Yelp Dataset Challenge `https://www.yelp.com/dataset/challenge`. To extract a cohesive subset of user-business ratings, we first select all businesses in the state of Arizona (the state with the largest number of ratings) with a selected set of restaurant categories. Further, we filter the data to retain only users and business which have at-least 15 ratings. Finally, we use a time-window of 6 months to extract 12 snapshots in the period of 2009 to 2015.

The ML-10m dynamic user-tag interaction network was downloaded from `http://networkrepository.com/ia-movielens-user2tags-10m.php`. This dataset depicts the tagging behavior of MovieLens users, with the tags applied by a user on her rated movies. We use a time-window of 3 months to extract 13 snaphots over the course of 3 years.

