# OpenReview forum: "Dynamic Graph Representation Learning via Self-Attention Networks"
_ICLR.cc/2019/Conference_

### Official Review · AnonReviewer2 · 2018-10-31
**Good paper, lacks comparison against a few key baselines.**

**Rating:** 5
**Confidence:** 4

**Review:**

This is a well-written paper studying the important problem of dynamic network embedding. Please find below some pros and cons of this paper.
Pros:

* Studies the important problem of network embedding under a more realistic setting (i.e., nodes & edges evolve over time).
* Introduces an interesting architecture that uses two forms of attention: structural and temporal.
* Demonstrated the effectiveness of the temporal layers through additional experiments (in appendix) and also introduced a variant of their proposed approach which can be trained incrementally using only the last snapshot.

Cons:

* The authors compared against several dynamic & static graph embedding approaches. If we disregard the proposed approach (DySAT), the static methods seem to match and even, in some cases, beat the dynamic approaches on the compared temporal graph datasets. The authors should compare against stronger baselines for static node embedding, particularly GAT which introduced the structural attention that DySAT uses to show that the modeling of temporal dependencies is necessary/useful. Please see [1] for an easy way to train GCN/GAT for link prediction.
* There are actually quite a number of work done on network embedding on dynamic graphs including [2-4]. In particular, [2-3] support node attributes as well as the addition/deletion of nodes & edges. The author should also compare against these work.
* The concept of temporal attention is quite interesting. However, the authors do not provide more analysis on this. For one, I am interested to see how the temporal attention weights are distributed. Are they focused on the more recent snapshots? If so, can we simply retain the more relevant recent information and train a static network embedding approach? Or are the attention weights distributed differently?

[1] Modeling Polypharmacy Side Effects with Graph Convolutional Networks. Zitnik et. al. BioInformatics 2018.
[2] Attributed Network Embedding for Learning in a Dynamic Environment. Li et. al. In Proc. CIKM '17.
[3] Streaming Link Prediction on Dynamic Attributed Networks. Li et. al. In Proc. WSDM '18.
[4] Continuous-Time Dynamic Network Embeddings. Nguyen et. al. In Comp. Proc. WWW '18.

---

> ### Author Response · Authors · 2018-11-26
> **Reply to AnonReviewer2 (Part 1)**
>
>
> We would like to thank you for the in-depth questions on our experimental results! Please refer to our global comment above for a list of all revisions to the paper -- we hope they have appropriately addressed your comments.
>
> We respond to each of your comments below as follows:
>
> Q1: The authors compared against several dynamic & static graph embedding approaches. If we disregard the proposed approach (DySAT), the static methods seem to match and even, in some cases, beat the dynamic approaches on the compared temporal graph datasets. The authors should compare against stronger baselines for static node embedding, particularly GAT which introduced the structural attention that DySAT uses to show that the modeling of temporal dependencies is necessary/useful. Please see [1] for an easy way to train GCN/GAT for link prediction.
>
> We agree with you on the observation that static methods often match or beat existing dynamic embedding methods. Our initial experiments contain comparison to GraphSAGE - an unsupervised representation learning framework that supports various neighborhood aggregation functions, including GCN and GAT aggregators.
>
> We thank you for the valuable pointer [1] which trains GCN/GAT as a graph autoencoder directly for link prediction. While conceptually similar to GCN/GAT variants of GraphSAGE, two key differences include (a) lack of neighborhood sampling, and (b) link prediction objective instead of random walk samples. To examine the effect of these differences on link prediction performance, we used the aforementioned implementation [1] to train autoencoder models of GCN and GAT, denoted by GCN-AE and GAT-AE in our experiments. Our experimental results have been updated to include these as static embedding methods for comparison. From the results, we find the performance of these methods to be mostly similar to their corresponding GraphSAGE variants, which is consistent with our expectation.

---

> > ### Author Response · Authors · 2018-11-26
> > **Reply to AnonReviewer2 (Part 2)**
> >
> >
> > Q2: There are actually quite a number of work done on network embedding on dynamic graphs including [2-4]. In particular, [2-3] support node attributes as well as the addition/deletion of nodes & edges. The author should also compare against these work.
> >
> > We thank you for the useful references on dynamic graphs. We are aware of these papers and do agree that they are related to our work. Consequently, we have revised Section 2 (Related work) in the revised paper to reflect the same. While we agree on the relevance of these works, we list below our reasons for not including experimental comparisons:
> >
> > >> Attributed Network Embedding for Learning in a Dynamic Environment. Li et. al. In Proc. CIKM '17 [2]:
> > This paper learns node embeddings in dynamic attributed graphs by initially training an offline model, followed by incremental updates over time.
> >
> > First, their key focus is online learning to improve efficiency over retraining static models, while our goal is to improve representation quality by capturing temporal evolutionary patterns in graph structure. This implies that their model can at best reach the performance of a statically re-trained method (as demonstrated in their paper), while we achieve significant improvements over static methods.
> >
> > Second, their proposed model DANE is designed for attributed graphs with evolving structure and attributes, while our model is designed for dynamic non-attributed graphs. A direct application of DANE to non-attributed graphs would not be optimal and indeed our initial experiments on using DANE indicate significantly inferior performance even versus simplest static embedding methods. Thus, to avoid an unfair comparison, we exclude DANE in our experimental results.
> >
> > >> Streaming Link Prediction on Dynamic Attributed Networks. Li et. al. In Proc. WSDM '18 [3]:
> > This paper focuses on link prediction in dynamic attributed graphs, but does not learn latent representations for nodes, hence directly orthogonal and does not address our problem of dynamic graph representation learning. We mention a few other differences to support our choice:
> >
> > First, they once again focus on online learning to enable scaling to large-scale streaming networks. Their key objective is on efficiency to support streaming graphs, while our goal is to learn latent node representations that capture evolutionary graph structures.
> >
> > Second, although their method might be relevant for comparison with our incremental variant IncSAT, we were unable to obtain the implementation even after contacting the authors. Since their method falls under the category of streaming graphs with a focus on efficiency and scalability, we believe an experimental comparison is outside the scope of our work.
> >
> > >> Continuous-Time Dynamic Network Embeddings. Nguyen et. al. In Comp. Proc. WWW '18 [4]:
> > This paper learns dynamic graph embeddings on temporal graphs with continuous time-stamped links.
> >
> > First, this paper operates under the assumption of continuous time-stamped links, which is often not realistic and distinct from the most established problem setup of using dynamic graph snapshots at discrete time steps. Thus, a direct comparison may not be fair.
> >
> > Second, this paper assumes a continuous-time dynamic graph with the restriction that each link occurs *only* once. This is an unrealistic assumption, which prevents applicability to  most real-world dynamic graphs including all of our considered datasets where each link typically occurs in multiple snapshots. Thus considering all these factors, we exclude a comparison with this method in our experiments.
> >
> > Q3: The concept of temporal attention is quite interesting. However, the authors do not provide more analysis on this. For one, I am interested to see how the temporal attention weights are distributed. Are they focused on the more recent snapshots? If so, can we simply retain the more relevant recent information and train a static network embedding approach? Or are the attention weights distributed differently?
> >
> > We thank you for your question on the analysis of temporal attention weights. We have conducted a preliminary study to analyze the distribution of attention weights learned by the temporal attention layers, over multiple time steps as requested. The results are reported in Appendix A.2. We choose the Enron dataset for this experiment, and present a heatmap of the normalized attentional coefficients with mean and standard deviation, over multiple time steps. Figure 3 in Appendix A.2 illustrates a mild bias towards recent snapshots while we observe significant variance in the attentional coefficients across different nodes in the graph.  This may indicate that the learned temporal attention weights capture historical context well, and vary to a considerable degree across different nodes in the graph.

---

### Official Review · AnonReviewer3 · 2018-11-02
**Dynamic Self-Attention Network**

**Rating:** 6
**Confidence:** 4

**Review:**

This paper proposes a model for learning node embedding vectors of dynamic graphs, whose edge topology may change. The proposed model, called Dynamic Self-Attention Network (DySAT), uses attention mechanism to represent the interaction of spatial neighbouring nodes, which is closely related to the Graph Attention Network. For the temporal dependency between successive graphs, DySAT also uses attention structure inspired by previous work in machine translation. Experiments on 4 datasets show that DySAT can improve the AUC of link prediction by significant margins, compared to static graph methods and other dynamic graph methods. Though the attention structures in this paper are not original, combining these structures and applying them on dynamic graph embedding is new.

Here are some questions:

1. What will happen if a never-seen node appears at t+1? The model design seems to be compatible with this case. The structural attention will still work, however, the temporal attention degenerates to a “static” result --- all the attention focus on the representation at t+1. I am curious about the model performance in this situation, since nodes may arise and vanish in real applications.

2. What is the performance of the proposed algorithm for multi-step forecasting? In the experiments, graph at t+1 is evaluated using the model trained up to graph_t. However, in real applications we may don’t have enough time to retrain the model at every time step. If we use the model trained up to graph_t to compute node embedding for the graph_{t+n}, what is the advantage of DySAT over static methods?

3. What is the running time for a single training process?

---

> ### Author Response · Authors · 2018-11-26
> **Reply to AnonReviewer3**
>
> We would like to thank you for the insightful questions on our experiments! Please refer to our global comment above for a list of all revisions to the paper.
>
> We respond to each of your comments below as follows:
>
> Q1: What will happen if a never-seen node appears at t+1? The model design seems to be compatible with this case. The structural attention will still work, however, the temporal attention degenerates to a “static” result - all the attention focus on the representation at t+1. I am curious about the model performance in this situation, since nodes may arise and vanish in real applications.
>
> We agree with the apt observation on the capability of DySAT to handle new nodes, and have conducted additional experiments to examine model performance in such situations. To compute the representation of a new node v at time step t+1, the only available information is the local structure around v at t+1. Although temporal attention will focus on the latest representation due to absence of history, it however does not degenerate to a “static” result. The temporal attention applied on the neighboring nodes of v (say N_v), indirectly contribute towards the embedding of v, through backpropagation updates. Specifically, the structural embedding of v is computed as a function of N_v, whose structural embeddings receive backpropagation signals through the temporal attention layers (assuming they are not all new nodes). Thus, temporal attention indirectly affects the final embedding of node v.
>
> As suggested, we empirically examine the performance of DySAT on “new” previously unseen nodes. We report link prediction performance *only* on the new nodes at each time step using the same experimental setup, i.e., a test example (node pair) is reported for evaluation only if it contains at least one new node. Due to the significant variance in the number of new nodes at each step, we report the performance (AUC) at each time step, along with a mention on the corresponding number of new nodes. The results are available in Figure 5 of the Appendix D. From Figure 5, we observe consistent gains of DySAT over other baselines, similar to our main results.
>
> Q2: What is the performance of the proposed algorithm for multi-step forecasting? In the experiments, graph at t+1 is evaluated using the model trained up to graph_t. However, in real applications we may don’t have enough time to retrain the model at every time step. If we use the model trained up to graph_t to compute node embedding for the graph_{t+n}, what is the advantage of DySAT over static methods?
>
> We agree with your view on the importance of not re-training the model at every time step in real-world applications. Multi-step forecasting is typically achieved either by (a) designing a model to predict multiple steps into the future, or by (b) recursively feeding next predictions as input, for a desired number of future steps. In case of dynamic graphs, events correspond to link occurrences, which renders forecasting different from conventional time-series, due to the occurrence of new nodes in each time step. Due to this key distinction, we list below, two possibilities for forecasting in dynamic graphs: (a) Link prediction at future step t+n (on all nodes) by incrementally updating the model on new snapshots. (b) Link prediction at future step t+n (among nodes present at t) based on dynamic embeddings learned at t, followed by a downstream classifier to predict the links at t+n. Note that (b) does not involve model re-training or updating while (a) requires incremental model updates.
>
> In our paper, we have examined (a) by proposing an incremental variant named IncSAT and report the performance in Table 5 of Appendix E.
>
> We have now added an additional experiment in Appendix C to evaluate forecasting using strategy (b), which enables direct evaluation of DySAT on multi-step link prediction. Here, each method is trained for a fixed number of time steps, and the latest embeddings are used to predict links at multiple future steps. In each dataset, we choose the last 6 snapshots to evaluate multi-step link prediction where we create examples from the links in G_{t+n} and an equal number of randomly sampled pairs of unconnected nodes (non-links). Our experimental results (Figure 4) indicate significant improvements for DySAT over all baselines and a highly stable link prediction performance over future time steps.
>
> Q3: What is the running time for a single training process?
>
> We have revised Section 5.4 to add the running time information. Specifically, we report the runtime of DySAT on a machine with Nvidia Tesla V100 GPU and 28 CPU cores. The runtime per mini-batch of DySAT with batch size of 256 nodes on the ML-10M dataset, is 0.72 seconds. In comparison, the model variant without the temporal attention (No Temporal) takes 0.51 seconds. Thus, structural attention constitutes a major fraction of the overall runtime, while the cost of temporal attention is relatively lower.

---

### Official Review · AnonReviewer1 · 2018-11-03
**Dynamic graph representation learning with self-attention**

**Rating:** 4
**Confidence:** 5

**Review:**

This paper describes learning representation for dynamic graphs using structural and temporal self-attention layers. They applied their method for the task of link-prediction. However, I have serious objections to their experimental setup. I have seen people used sets of edges and pairs of vertices without an edge for creating examples for link-prediction on a static graph, however, working with a real-world dynamic graph, you can compute the difference between G_t and G_{t+1} as the changes that occur in G_t+1 1) Why are you not trying to predict these changes?  Moreover, 2) why do you need examples from snapshot t+1 for training when you have already observed t snapshots of the graph?
3) The selected graphs are very small comparing to the dynamic graphs available here http://konect.uni-koblenz.de/networks/.

---

> ### Author Response · Authors · 2018-11-26
> **Reply to AnonReviewer1 (Part 1)**
>
> We would like to thank you for your review with thoughtful questions on the experiments! Please refer to our global comment above for a list of all revisions to the paper.
>
> We respond to each of your comments below as follows:
>
> Q1: I have seen people used sets of edges and pairs of vertices without an edge for creating examples for link-prediction on a static graph, however, working with a real-world dynamic graph, you can compute the difference between G_t and G_{t+1} as the changes that occur in G_{t+1} 1) Why are you not trying to predict these changes?
>
> We agree with the observation that the differences between graphs G_t and G_{t+1} can be computed in real-world dynamic graphs, and we have included additional experiments on new link prediction in Appendix B.
>
> First, we would like to clarify our view of dynamic link prediction based on our understanding of existing literature. The goal of dynamic link prediction is to predict the set of future links (or interactions) based on historically observed graph snapshots. In practice, this can be realized as predicting future user interactions in email communication networks or user-item ratings in recommender systems. In such scenarios, dynamic link prediction aims to predict the set of “all” future links (at time step t+1) given history until time step t. To the best of our knowledge, this evaluation approach has been widely adopted in our surveyed literature on dynamic link prediction in Section 2 (Related Work) of the paper. Our compared dynamic graph embedding baselines DynamicTriad, DynGEM, and Know-Evolve also adopt the same convention, by evaluating the predicted links at (t+1) through classification and ranking metrics.
>
> On the other hand, we do agree with your perspective that a dynamic graph representation should be evaluated in its ability to predict “new” links. We have added an additional experiment (Appendix B) where evaluation examples comprise “new” links at G_{t+1} (which have not been observed in G_t), and an equal number of randomly sampled pairs of unconnected nodes (non-links). We use a similar evaluation methodology to evaluate the performance of dynamic link prediction through AUC scores. This experiment specifically evaluates the ability of different methods to predict new links at (t+1).
>
> Though the overall prediction accuracies are generally lower in comparison to the previous experimental setting, we observe consistent gains of 3-5% over the best baseline similar to our earlier results. The new results can be found in Table 4 of Appendix B, along with accompanying discussion. We hope that the addition of this experiment further showcases the capability of DySAT for dynamic link prediction.
>
> Q2: Why do you need examples from snapshot t+1 for training when you have already observed t snapshots of the graph?
>
> Firstly, the training step for all models only utilizes the snapshots up to t to compute the embeddings for all nodes, which can subsequently be used in different downstream tasks such as link prediction, classification, clustering, etc. No data from snapshot t+1 are utilized in training the node embedding model. Since we focus on dynamic link prediction as the primary task for evaluation, the goal to predict future links (at time step t+1) given history until time step t. Thus, the evaluation set consists of examples from snapshot t+1.
> Secondly, the examples from time snapshot t+1 are *only* used to train a downstream logistic regression classifier for evaluating link prediction performance. Since the evaluation set comprises the links at t+1, we choose a small percentage of those examples (20%) for training, which is consistent with standard evaluation procedures. We follow the same setup for all the compared methods. In case of a different task such as multi-step forecasting to predict links at t+n, we similarly use 20% of examples at t+n for training the downstream classifier. We have revised the draft to make the experiment setup clearer.
>
> Meanwhile, we also describe the reason for using a downstream classifier to evaluate link prediction. Arguably, link prediction can also be evaluated by applying a sigmoid function on the inner product computed on pairs of node embeddings at time step t. However, we instead choose to train a downstream classifier (as done in node2vec, DynamicTriad etc.) to provide a fair comparison against baselines (such as DynamicTriad), which use other distance metrics (L_1 distance, etc.) for link prediction. We believe this evaluation methodology provides a more flexible framework to fairly evaluate various methods which are trained using different distance/proximity metrics.

---

> > ### Author Response · Authors · 2018-11-26
> > **Reply to AnonReviewer1 (Part 2)**
> >
> > Q3: The selected graphs are very small comparing to the dynamic graphs available here http://konect.uni-koblenz.de/networks/.
> >
> > We thank the reviewer for the useful pointer to an extensive collection of real-world dynamic graphs.
> >
> > First, our experiments are conducted on real-world communication and rating networks with over 20,000 nodes and nearly 100,000 edges, which we believe constitute a diverse and representative sample of real-world dynamic graphs. Due to lack of established benchmark datasets for dynamic graphs, we choose Enron, UCI, Yelp, and MovieLens which have been widely in analysis of dynamic graphs.
> >
> > Second, among the 7 compared baseline models on graph representation learning, 5 of them (GCN, GAT, node2vec, GraphSAGE, and DynGEM) choose their experiment datasets with comparable or smaller sizes. As mentioned in Section 6, our current implementation requires storing the sparse adjacency matrices of each snapshot in GPU memory, which limits scaling to graphs with over millions of nodes. This is a common issue faced by many successful graph neural network architectures such as GCN, GAT, etc.
> >
> > Since DySAT builds on the same framework as GCN and GAT, we foresee a direct extension to incorporate efficient neighborhood sampling strategies (similar to GraphSAGE and others), thus scaling to larger-scale dynamic graphs and we leave it as our future work.
> >
> > Finally, we would like to point out that the sizes of graph used in our experiments are comparable and often larger than the widely established benchmark citation networks Cora, Citeseer, and, Pubmed datasets for node classification in static graphs.

---

### Public Comment · ~Michael_Bronstein1 · 2018-09-30
**many prior works missing**

I would like to draw the authors' attention to multiple recent works on deep learning on graphs directly related to their work. Among spectral-domain methods, the fundamental work of Bruna et al. [1] has started the recent interest in deep learning on graphs. Replacing the explicit computation of the Laplacian eigenbasis of the spectral CNNs in [1] with polynomial [2] and rational [3] filter functions is a very popular approach (the cited method of Kipf&Welling is a particular setting of [1]). On the other hand, there are several spatial-domain methods that generalize the notion of patches on graphs. These methods originate from works on deep learning on manifolds in computer graphics and recently applied to graphs, e.g. the Mixture Model Networks (MoNet) [4] (Note that the cited Graph Attention Networks (GAT) of Veličković et al. are a particular setting of [4]). MoNet architecture was generalized in [5] using more general learnable local operators and dynamic graph updates. A further generalization of GAT is the dual graph attention mechanism [6]. Finally, the authors may refer to a review paper [7] on non-Euclidean deep learning methods.


1. Spectral Networks and Locally Connected Networks on Graphs, arXiv:1312.6203.

2. Convolutional Neural Networks on Graphs with Fast Localized Spectral Filtering, arXiv:1606.09375

3. CayleyNets: Graph convolutional neural networks with complex rational spectral filters, arXiv:1705.07664,

4. Geometric deep learning on graphs and manifolds using mixture model CNNs, CVPR 2017.

5. Dynamic Graph CNN for learning on point clouds, arXiv:1712.00268

6. Dual-Primal Graph Convolutional Networks, arXiv:1806.00770.

7. Geometric deep learning: going beyond Euclidean data, IEEE Signal Processing Magazine, 34(4):18-42, 2017

---

> ### Author Response · Authors · 2018-09-30
> **re: many prior works missing**
>
> Thank you very much for providing a detailed description of prior work on deep learning on graphs.
> We are aware of most of the works that you have mentioned, which fall into the category of static graph representation learning. Due to our focus on the dynamic graph setting and limited space, we limit our attention mainly on most recent and related state-of-the-art works GCN (Kipf & Welling), GraphSAGE (Hamilton et al.) and GAT (Veličković et al.). However, we agree that the mentioned papers are relevant and we will be sure to cite and discuss them in a subsequent version of our paper.

---

### Author Response · Authors · 2018-11-26
**Summary of revisions**


We hope our revisions to the paper have adequately addressed the comments of the reviewers, and other interested anonymous researchers.  We would like to thank everyone for their insightful comments on our paper and sincerely believe that it has helped improve the overall quality and contribution.

* We have added new experiments on dynamic new link prediction, to compare different graph representation learning methods on link prediction focused on (a) new links, and (b) previously unseen new nodes. Our results (summarized in Appendices B and D) indicate similar performance improvements for DySAT over existing methods.

* We have conducted additional experiments to evaluate DySAT on multi-step link prediction/forecasting (Appendix C) where the node embeddings trained until G_t, are used to predict the links at future snapshot G_{t+n}. DySAT achieves significant improvements over all baselines and maintains a highly stable link prediction performance over future time steps.

* In all of our experimental results, we additionally compare against two static graph embedding baselines, GCN and GAT trained for link prediction, denoted by GCN-AE and GAT-AE respectively. As expected, their performance is typically close to the corresponding GraphSAGE variants.

* We have added an experiment in Appendix A.2 to visualize the distribution of attention weights learned by the temporal attention layers, over multiple time steps. Our results indicate a mild bias towards recent snapshots while exhibiting significant variability across different nodes in the graph.

* In Section 2, we have added appropriate references to discuss relevant related work on dynamic attributed graphs and continuous-time dynamic graphs.

* In section 5.4, we have added details on the running time of DySAT, comparing the relative costs of structural and temporal attention.

---

### Author Response · Authors · 2018-12-03
**Feedback after revisions**


Dear Reviewers and ACs,

We thank you once again for the time and effort to review our paper, and appreciate the valuable questions and suggestions. We have made several improvements to our paper, and hope that they sufficiently address your key concerns. We would greatly value any additional comments on the revised paper, for further discussion and improvement.

---

### Meta-Review · Area_Chair1 · 2018-12-14
**Novel application of self attention for estimating dynamic graph embeddings**

**Confidence:** 4
**Recommendation:** Reject

**Metareview:**

This paper proposes a self-attention based approach for learning representations for the vertices of a dynamic graph, where the topology of the edges may change. The attention focuses on representing the interaction of vertices that have connections. Experimental results for the link prediction task on multiple datasets demonstrate the benefits of the approach. The idea of attention or its computation is not novel, however its application for estimating embeddings for dynamic graph vertices is new.
The original version of the paper did not have strong baselines as noted by multiple reviewers, but the paper was  revised during the review period. However, some of these suggestions, for example, experiments with larger graph sizes and other related work i.e., similar work on static graphs are left as a future work.